# Co-Application of TiO_2_ Nanoparticles and Arbuscular Mycorrhizal Fungi Improves Essential Oil Quantity and Quality of Sage (*Salvia officinalis* L.) in Drought Stress Conditions

**DOI:** 10.3390/plants11131659

**Published:** 2022-06-23

**Authors:** Ali Ostadi, Abdollah Javanmard, Mostafa Amani Machiani, Amir Sadeghpour, Filippo Maggi, Mojtaba Nouraein, Mohammad Reza Morshedloo, Christophe Hano, Jose M. Lorenzo

**Affiliations:** 1Department of Plant Production and Genetics, Faculty of Agriculture, University of Maragheh, Maragheh P.O. Box 55136-553, Iran; aliostadi1369@gmail.com (A.O.); amani0056@gmail.com (M.A.M.); mojtabanouraein@yahoo.com (M.N.); 2Crop, Soil, and Environment Program, School of Agricultural Sciences, Southern Illinois University of Carbondale, College of Science, Carbondale, IL 62901, USA; amir.sadeghpour@siu.edu; 3School of Pharmacy, Chemistry Interdisciplinary Project (ChIP), University of Camerino, 62032 Camerino, Italy; filippo.maggi@unicam.it; 4Department of Horticultural Science, Faculty of Agriculture, University of Maragheh, Maragheh P.O. Box 55136-553, Iran; morshedlooreza@gmail.com; 5Laboratoire de Biologie des Ligneux et des Grandes Cultures, INRA USC1328, Orleans University, CEDEX 2, 45067 Orléans, France; 6Centro Tecnológico de la Carne de Galicia, Rúa Galicia Nº 4, Parque Tecnológico de Galicia, San Cibraodas Viñas, 32900 Ourense, Spain; jmlorenzo@ceteca.net; 7Área de Tecnología de los Alimentos, Facultad de Ciencias de Ourense, Universidad de Vigo, 32004 Ourense, Spain

**Keywords:** medicinal and aromatic plants, secondary metabolites, sustainable agriculture, thujone, water deficit

## Abstract

Drought stress is known as a major yield-limiting factor in crop production that threatens food security worldwide. Arbuscular mycorrhizal fungi (AMF) and titanium dioxide (TiO_2_) have shown to alleviate the effects of drought stress on plants, but information regarding their co-addition to minimize the effects of drought stress on plants is scant. Here, a two-year field experiment was conducted in 2019 and 2020 to evaluate the influence of different irrigation regimes and fertilizer sources on the EO quantity and quality of sage (*Salvia officinalis* L.). The experiment was laid out as a split plot arranged in a randomized complete block design with three replicates. The irrigation treatments were 25, 50, and 75% maximum allowable depletion (MAD) percentage of the soil available water as non-stress (MAD_25_), moderate (MAD_50_), and severe (MAD_75_) water stress, respectively. Subplots were four fertilizer sources including no-fertilizer control, TiO_2_ nanoparticles (100 mg L^−1^), AMF inoculation, and co-addition of TiO_2_ and AMF (TiO_2_ + AMF). Moderate and severe drought stress decreased sage dry matter yield (DMY) by 30 and 65%, respectively. In contrast, application of TiO_2_ + AMF increased DMY and water use efficiency (WUE) by 35 and 35%, respectively, compared to the unfertilized treatment. The highest EO content (1.483%), yield (2.52 g m^−2^), and *cis*-thujone (35.84%, main EO constituent of sage) was obtained in MAD_50_ fertilized with TiO_2_ + AMF. In addition, the net income index increased by 44, 47, and 76% with application of TiO_2_ nanoparticles, AMF, and co-addition of TiO_2_ + AMF, respectively. Overall, the integrative application of the biofertilizer and nanoparticles (TiO_2_ + AMF) can be recommended as a sustainable strategy for increasing net income and improving EO productivity and quality of sage plants in drought stress conditions. Future policy discussions should focus on incentivizing growers for replacing synthetic fertilizers with proven nano and biofertilizers to reduce environmental footprints and enhance the sustainability of sage production, especially in drought conditions.

## 1. Introduction

Industrial demand for the production and use of medicinal and aromatic plants and their byproducts has increased due to negative effects of synthetic drugs. Sage (*Salvia officinalis* L.) is a perennial medicinal and aromatic plant which belongs to the Lamiaceae family and is widely used in the pharmaceutical and food industries. The global production of sage in the world is estimated to be about 50–100 tons per year [1]. Sage contains therapeutically effective compounds and has been used in the treatment of 60 diseases such as skin diseases, bronchitis, mouth and throat inflammations, digestive and circulation disturbances, cough, and other diseases [2]. The sage essential oils (EOs) can provide antioxidant, antimicrobial, antimutagenic, anticholinesterase, anti-inflammatory, and antibiotic benefits [3,4]. The major components are *cis*-thujone, camphor, 1,8-cineole, *trans*-thujone, viridiflorol, and camphene [5].

In the last century, the average temperature of the Earth has increased by 0.7 °C, leading to the transformation of temperate climates into semi-arid and arid climates [6]. More than 88% of farmlands in Iran are in arid and semi-arid regions [7]. The average annual rainfall in Iran is around 250 mm per year, which is approximately 75% lower than the global average rainfall (990 mm per year) [8]. Drought is a major abiotic stress that adversely impacts the morphology and physiology of plants. Drought stress negatively affects nutrient absorption, photosynthetic capacity, and water use efficiency (WUE) of plants; therefore, it decreases plant productivity and quality [9,10,11]. The decreasing plant productivity under drought conditions has become an important challenge for agricultural producers to ensure the food security in arid and semi-arid regions [12]. In addition, the lower nutrient use efficiency in drought stress conditions forces farmers to use more chemical inputs. A large portion of chemical fertilizers are lost and also become unavailable to plants in drought conditions. In general, 40–70% of N, 80–90% of P, and 50–90% of K are lost and/or fixed in soils as a result of drought, which has huge economic consequences [13,14]. Two strategies to alleviate drought stress effects on plant production and quality are represented by harnessing advances in nanotechnology such as the application of TiO_2_ and inoculating plants with biofertilizers such as arbuscular mycorrhizal fungi (AMF).

Nanotechnology is a modern strategy that has shown to increase the tolerance of agricultural crops to cope with drought stress [15]. Nanoparticles are tiny materials having size ranges from 1 to 100 nm. These particles have exclusive physical and chemical properties including high surface area, maximum uptake, high reactivity, sustainability, and less pollution [16,17]. Among different nanoparticles, TiO_2_ nanoparticles have shown to increase nutrient uptake, improve chlorophyll content, and promote light capture in chlorophylls (a and b). They also improve regulation of important enzyme activities such as glutamine synthase, glutamate dehydrogenase, and also nitrate reductase that helps the plants to absorb nitrate and also favors the conversion of inorganic N to organic N. Other benefits of TiO_2_ include an increase in N fixation and electron transfer activities, and improving carbon dioxide (CO_2_) assimilation and photosynthetic activities [18,19]. In addition, TiO_2_, as an anti-stress agent, decreases the negative effects of environmental stress by increasing the antioxidant enzyme activity [20]. Previous studies have reported the positive effects of TiO_2_ nanoparticles on the improvement of plant performance, especially medicinal and aromatic plants, in drought conditions. Ahmad et al. [21] noted that the application of TiO_2_ nanoparticles increased the EO quantity and yield by 39 and 105%, respectively, in peppermint seedlings. Gohari et al. [20] reported that the application of TiO_2_ nanoparticles improved EO quantity and quality of *Dracocephalum moldavica* L. in drought conditions. Additionally, Khater et al. [22] indicated that the foliar application of TiO_2_ nanoparticles improved EO quality of coriander (*Coriandrum sativum* L.) through increasing the main EO constituents such as linalool. Shabbir et al. [23] reported that the application of TiO_2_ nanoparticles enhanced dry matter yield (DMY) of vetiver (*Vetiveria zizanioides* L. Nash) as well as physiological characteristics such as total chlorophyll content, net photosynthetic rate, intercellular CO_2_ concentration, and carbonic anhydrase and nitrate reductase activity. Furthermore, the authors noted that the EO content and yield as well as khusimol (main active constituent of EO) content increased by 23.6, 55.1, and 24.5%, respectively, with the application of TiO_2_ nanoparticles.

Another promising and effective strategy to cope with drought stress is biofertilization or inoculation of crops with AMF. Arbuscular mycorrhizal fungi are soil-borne fungi which can significantly improve plant performance by increasing nutrient absorption and resistance to stressful conditions such as drought, salinity, heat, and heavy metals. AMF-plants’ roots symbiotic relationship enhances nutrient and water uptake and regulates plant physiological functions including leaf water potential, relative water content, stomatal conductance, and CO_2_ stabilization [24,25,26]. Previous studies have shown that inoculating plants’ roots with AMF increases the EO quantity and quality of medicinal and aromatic plants such as thyme (*Thymus vulgaris* L.) [27], peppermint (*Mentha* x *piperita* L.) [28], and holy basil (*Ocimum tenuiflorum* L.) [29].

Integrating multiple strategies has proven to be more effective in minimizing the effects of different stress conditions in crop production and its quality [30]. The literature is scant on the potential of a TiO_2_ and AMF addition to alleviate drought stress in sage production systems. Therefore, the study aimed to investigate the effectiveness of treatment with AMF and TiO_2_ nanoparticles on sage morphology, physiology, yield, and EO quality under drought conditions, and investigate the economic incomes of each management practice to find the most environmentally friendly and economically viable options for growers. We hypothesized that the co-addition of TiO_2_ nanoparticles and AMF would be a sustainable strategy to decrease the effects of drought and increase the EO quality in sage.

## 2. Results

### 2.1. Arbuscular Mycorrhizal Fungi (AMF) Colonization

The AMF colonization percentage was significantly influenced by irrigation regimes, fertilizer, and irrigation regimes × fertilizer interaction. The maximum AMF colonization percentage was recorded from MAD_25_ irrigation levels inoculated with AMF (83.83%) followed by the TiO_2_ + AMF (81.50%) treatment. The AMF colonization was reduced by increasing the drought stress levels. However, the AMF colonization increased in drought stress conditions (MAD_50_ and MAD_75_) in AMF and TiO_2_ + AMF treatments (Figure 1).

### 2.2. Nutrient Concentration

ANOVA results showed that concentrations of N, P, and K in sage were significantly affected by irrigation regimes, fertilizer, and irrigation regimes × fertilizer (Table 1). The maximum concentration of N (29.9 g kg^−1^), P (2.20 g kg^−1^), and K (29.73 g kg^−1^) was obtained in the normal irrigation regime (MAD_25_) fertilized with TiO_2_ + AMF. However, the lowest content of N (21.0 g kg^−1^), P (1.47 g kg^−1^), and K (19.6 g kg^−1^) was obtained under severe drought stress (MAD_75_) without fertilization, reflecting the drought impact on mechanisms by which plants acquire macro-nutrients (Table 1).

Our results indicated that in no-drought (MAD_25_) and moderate (MAD_50_) stress conditions, the co-addition of TiO_2_ + AMF significantly increased N concentrations. However, either fertilization with TiO_2_ or inoculation with AMF improved N acquisition by sage like the co-addition of the two sources (Table 1).

Except for in the optimal condition (MAD_25_), AMF inoculation increased the P concentration of sage similar to or higher than the co-addition of TiO_2_ + AMF, indicating that AMF are efficient in delivering P in moderate (MAD_50_) and severe (MAD_75_) drought conditions.

Application of TiO_2_ was ineffective in increasing the K concentration of sage in moderate (MAD_50_) and severe (MAD_75_) drought conditions compared to AMF inoculation and co-addition of TiO_2_ + AMF, indicating that AMF are effective in providing macro-nutrients in drought stress conditions. The content of N, P, and K decreased by 12, 13, and 9% under moderate stress and 21, 20.9, and 21% under severe drought stress, respectively (Table 1).

### 2.3. Plant Height

The plant height of sage was influenced by irrigation regimes and fertilizer but not by their interaction. The plant height of sage was 41.74 cm in no-stress (MAD_25_) conditions, which was 14.2 and 30.6% higher than that in moderate (MAD_50_) and severe (MAD_75_) drought stress treatments, respectively (Figure 2a). The lowest plant height was recorded in the no-fertilizer control (Figure 2b). Addition of TiO_2_ and AMF similarly increased the plant height of sage. Co-addition of TiO_2_ and AMF increased the plant height compared with the TiO_2_ treatment but not AMF, indicating that AMF inoculation was enough to maximize the plant height (Figure 2b).

### 2.4. Canopy Diameter

The canopy diameter of sage was affected by irrigation regimes and fertilizer but not by their interaction. As drought stress increased, sage diameter decreased from 38.55 cm in the no-stress control to 33.10 cm in moderate (MAD_50_) and 23.90 cm in severe (MAD_75_) drought stress treatments (Figure 3a). In comparison with MAD_25_, the plant height and canopy diameter reduced by 16.2 and 14.2% in MAD_50_ and 30.6 and 38% in MAD_75_, respectively (Figure 2a and Figure 3a). Among different fertilizer sources, the highest and lowest canopy diameter of sage were measured with the co-addition of TiO_2_ + AMF and in the no-fertilizer control (Figure 3b). Compared with the no-fertilizer control, the application of TiO_2_, inoculation with AMF, and co-addition of TiO_2_ + AMF increased the canopy diameter by 24, 16, and 28%, respectively (Figure 3b). These results suggest that the co-addition of TiO_2_ + AMF provides a synergistic effect on plant morphology (Figure 2b and Figure 3b).

### 2.5. Dry Matter Yield

The dry matter yield of sage was influenced by irrigation regimes, fertilizer, and irrigation regimes × fertilizer interaction (Table 2). The maximum (262.85 g m^−2^) and minimum DMY (62.75 g m^−2^) were recorded in the no-stress control (MAD_25_) treatment fertilized with TiO_2_ + AMF and in the severe drought stress treatment (MAD_75_) without fertilization (Table 2). While the co-addition of AMF and TiO_2_ significantly increased the DMY of sage in the no-stress control (MAD_25_), it was ineffective in increasing sage DMY under moderate and severe drought stress, indicating that these amendments are most effective at optimum irrigation and that drought impacts their effectiveness.

### 2.6. Water Use Efficiency (WUE)

Water use efficiency was significantly affected by different irrigation regimes, fertilizer, and interaction of irrigation regimes × fertilizer (Table 2). The maximum WUE (226.83 g m^−3^) was recorded from MAD_25_ with the integrative application of TiO_2_ + AMF. In contrast, MAD_75_ without fertilization had the lowest WUE (108.35 g m^−3^) (Table 2). In the no-stress treatment, the co-addition of TiO_2_ + AMF was required to increase the WUE compared with other treatments. The addition of TiO_2_ or AMF never increased the WUE compared with the no-fertilizer control, regardless of irrigation treatment, indicating that TiO_2_ fertilization and AMF inoculation alone are not effective strategies to increase WUE.

### 2.7. Essential Oil Content and Yield

Irrigation regimes, fertilizer, and irrigation regimes × fertilizer and irrigation regimes × year interactions had a significant impact on the sage EO content (Table 2). The highest EO content of sage was recorded in the moderate drought stress (MAD_50_) condition fertilized with either TiO_2_ (1.410%) or TiO_2_ + AMF (1.483%) (Table 2). Additionally, the highest (2.52 g m^−2^) and lowest EO yield (0.56 g m^−2^) were obtained in MAD_50_ fertilized with TiO_2_ + AMF and MAD_75_ without fertilization. This indicates that moderate drought stress positively improves the EO content of sage and TiO_2_ fertilization is an effective strategy to further improve it. The essential oil content was the lowest at the optimal irrigation level (MAD_25_) without fertilization. The optimal irrigation level, fertilization with TiO_2_, or inoculation with AMF increased the EO content, indicating that both TiO_2_ and AMF are effective strategies for improving this parameter in sage. Under severe drought stress (MAD_75_), fertilization did not increase the EO content of sage, indicating that drought impacts the effectiveness of these fertilizer sources (Table 2).

### 2.8. Essential Oil Composition

In aerial parts of sage plants, the GC-MS-based analysis detected 33 compounds (94.37–98.71% of total composition). The main compounds were cis-thujone (30.73–35.84%), camphor (14.58–18.61%), 1,8-cineole (9.17–10.71%), trans-thujone (4.59–5.68), and camphene (3.11–4.35%). The maximum content of cis-thujone and camphene was obtained in MAD_50_ with application of TiO_2_ + AMF. Furthermore, the highest content of 1,8-cineole was achieved in MAD_75_ fertilized with TiO_2_. Additionally, the highest content of camphor was obtained in MAD_75_ treated with TiO_2_ + AMF. The content of cis-thujone, trans-thujone, and 1,8-cineole was enhanced by 7, 9.9, and 4.8% with the application of TiO_2_ + AMF, respectively (Table 3).

### 2.9. Chlorophyll Concentration

The concentration of chlorophyll a, b, and total were significantly affected by different irrigation regimes, fertilizer, and irrigation regimes × fertilizer interaction (Table 4). The co-application of TiO_2_ + AMF in the optimal irrigation treatment (MAD_25_) had the highest concentration of chlorophyll a (5.56 mg g^−1^ fresh weight), b (1.30 mg g^−1^ fresh weight), and total (6.85 mg g^−1^ fresh weight). In contrast, the lowest concentrations of chlorophylls were observed in the severe drought stress (MAD_75_) condition without fertilization (Table 4). The concentration of chlorophyll a, b, and total decreased by 34, 8, and 27% in MAD_50_ and 68, 72, and 68% in MAD_75_, respectively. Moreover, the concentration of the three mentioned chlorophylls were enhanced by 45, 35, and 43% with application of AMF + TiO_2_, respectively (Table 4).

### 2.10. Carotenoid Concentration

Different irrigation regimes, fertilizer, and irrigation regimes × fertilizer interaction had a significant effect on the carotenoid concentration (Table 4). The highest (1.90 mg g^−1^ fresh weight) carotenoid concentration in sage leaves was recorded in the moderate drought stress (MAD_50_) treatment with AMF inoculation. The lowest carotenoid concentration (0.44 mg g^−1^ fresh weight) in sage leaves was recorded in the severe drought stress (MAD_75_) treatment without fertilization. Inoculating sage with AMF only increased the carotenoid concentrations in moderate drought conditions and was ineffective in both the no-stress and severe drought stress condition (Table 4).

### 2.11. Relative Water Content (RWC)

The relative water content of sage leaves was only influenced by irrigation regimes and fertilizer source and not the interaction of irrigation regimes × fertilizer. The maximum (88.16%) RWC was recorded from MAD_25_, which was 13 and 34.8% higher than those of MAD_50_ and MAD_75_, respectively (Figure 4a). Among different fertilizer sources, the highest RWC content (82.98%) of sage leaves was measured in the AMF treatment, which was similar to the co-addition of TiO_2_ + AMF (80.48%) but 10.1 and 15.3% higher than that of the TiO_2_ addition and the no-fertilizer control, respectively (Figure 4b).

### 2.12. Net Income

In the optimal irrigation treatment (MAD_25_), AMF inoculation or the co-addition of TiO_2_ + AMF were the most economical decisions for sage growers. In a moderate drought condition (MAD_50_), fertilization with TiO_2_ or the co-addition of TiO_2_ + AMF were effective strategies to increase sage growers’ income. In a severe drought condition (MAD_75_), none of the strategies increased growers’ net income index (Figure 5). This indicates that either recent soil amendment technologies should become less expensive or there is a need for government subsidies to allow for the use of these soil amendments in severe drought conditions.

### 2.13. Correlation

Correlation results showed that DMY had a positive and significant correlation with all studied traits except for EO content and constituents. Moreover, chlorophyll a was positively correlated with chlorophyll b, chlorophyll total, carotenoids, WUE, N, P, and K concentrations (r = 0.73, 0.99, 0.52, 0.67, 0.78, 0.46 and 0.57, respectively). Additionally, the EO content showed significant correlations with EO yield, chlorophyll b, carotenoids, WUE, and cis-thujone (r = 0.58, 0.38, 0.30, 0.29, and 0.40, respectively). cis-Thujone showed a significant correlation with 1,8-cineole (r = 0.50, *p* value < 0.01). In addition, the content of camphor had a negative correlation with 1,8-cineole and cis-thujone (r = −0.34 and −0.56, respectively, *p* value < 0.01) (Figure 6).

## 3. Discussion

This study aimed to investigate the effect of AMF biofertilizer and TiO_2_ nanoparticles on growth, physiology, yield, and EO and nutrient concentration of sage in normal versus drought (moderate and severe) conditions. The results indicated that drought stress decreased AMF colonization, which was attributed to the degeneration of population and soil microbial activities as well as decreasing spore germination, spore density, root exudates, and supply carbohydrates by the host plant [31]. Similarly, Amani Machiani et al. [27] reported that the AMF colonization rate significantly reduced in drought stress conditions.

The results also indicated that the plant growth parameters including plant height, canopy diameter, and DMY significantly decreased in drought stress conditions. The loss of plant productivity in limited water conditions could be attributed to the impairment of the cell differentiation, division, and cell elongation rate through declining turgor pressure. Additionally, the photosynthetic rate of plants in drought stress conditions decreased due to the closing of stomata and reduced uptake of CO_2_ [32]. In addition, the decrease in plant performance was due to the lower nutrient availability as a result of the drought impact on the mass flow of N and diffusion of P and K into the plant roots [33]. Similarly, Amani Machiani et al. [27] reported that the thyme (*Thymus vulgaris* L.) DMY reduced with moderate and severe water scarcity due to the decrease in macro- and micro-nutrients uptake in drought conditions.

In contrast, the integrative application of TiO_2_ nanoparticles with AMF biofertilizer (TiO_2_ + AMF) enhanced the plant productivity in drought stress conditions. In this study, the concentration of macro-nutrients on the sage leaves increased with fertilization with TiO_2_ + AMF. Therefore, the higher productivity of sage seedlings with the integrative application of TiO_2_ + AMF could be explained by the role of the mentioned fertilizers in increasing the nutrient availability, which leads to optimal plant growth performance. In addition, the expansion of the extra-radical mycelium and the increase in the surface absorbing capability of host roots can effectively enhance the absorption of nutrients and improve plant yield [34]. In addition, the higher nutrient uptake by inoculation of AMF could be due to the increasing soil acidity around the rhizosphere by the release of H^+^ ions, leading to solubility of macro- and micro-nutrients [35]. Furthermore, TiO_2_ enhances nutrient accessibility by regulating enzyme activity involved in N metabolisms including nitrate reductase, glutamine synthase, etc. [36]. It seems that the integrative application of these biofertilizers and nanoparticles enhances the sage productivity as a result of the increase in nutrient uptake, photosynthesis rate, and stocks of carbohydrates in plant organs and their transfer (from sinks to sources) during the reproductive stage.

Essential oils are used in the food and pharmaceutical industries due to their characteristic flavor and fragrance properties and some biological activities. The results of this study demonstrated that the sage EO content and components’ percentages were enhanced under moderate drought stress (MAD_50_). The increase in EO content under drought stress conditions is one of the main defense mechanisms in medicinal and aromatic plants. In this condition, the photosynthetic rate decreased due to the reduction in CO_2_ uptake (as a result of closing stomata) leading to the generation of NADPH+ H^+^ [26]. The increase in NADPH+ H^+^ is known as an inhibitory factor for photosynthetic rate and plant productivity. Therefore, the productivity of secondary metabolites such as EO compounds, alkaloids, etc., as a function of the increase in NADPH+ H^+^ consumption (accumulated in plant cells) can effectively decrease the negative impacts of drought stress in medicinal and aromatic plants [37,38]. Similarly, Govahi et al. [39] noted that the maximum EO content of sage was enhanced by 109 and 84% in moderate and severe drought stress conditions, respectively.

Furthermore, the fertilization of plants with TiO_2_ + AMF significantly increased the EO contents and main constituents’ percentages. It seems that the integrative application of TiO_2_ + AMF improves the nutrient accessibility as a result of stronger seedlings and higher metabolic efficiency (e.g., nutrient transport), which have an important role in the production of carbohydrates and the development of the glandular trichomes, EO channels, and secretory ducts. Additionally, TiO_2_ nanoparticles help to convert inorganic N into organic N in the form of protein and chlorophyll, which leads to an increase in the photosynthetic rate [40]. Zhao et al. [41] noted that the increase in primary photosynthesis compounds including erythrose-4-phosphate, phosphoenolpyruvate, pyruvate, and glyceraldehyde-3-phosphate plays a main role in increasing terpene constituents and EO productivity in medicinal and aromatic plants. In addition, Hazzoumi et al. [42] concluded that AMF inoculation stimulates the productivity of EO glands probably by increasing endogenous hormone levels, particularly cytokinin, indole-3-acetic acid, and gibberellin.

The EO yield is calculated by multiplying the DMY with the EO content of plants and has a positive correlation with the two mentioned factors (Figure 6). Therefore, the enhancement of EO yield in MAD_50_ was related to increasing EO productivity in moderate stress conditions and the effect of the TiO_2_ + AMF application in increasing the DMY and EO content.

Chlorophylls play an important role in light-harvesting, energy transfer, and electron transfer in photosynthetic organisms. The decrease in chlorophyll content in water-limiting conditions may be due to the increase in lipid peroxidation on the membranes as a result of the accumulation of reactive oxygen species (ROS), leading to chloroplast breakdown, a decrease in chlorophyllase activity, and the degradation of chlorophyll precursors [43]. However, the carotenoid content increased in moderate drought stress (MAD_50_). The increase in carotenoid content is one of the non-enzymatic defense mechanisms for increasing plant performance [44]. In drought stress conditions, carotenoids have antioxidant, anti-aging, and ROS scavenging activities in plant cells [27,45]. Therefore, the increase in carotenoids reduces the negative effects of drought stress by inhibiting ROS production and lipid peroxidation. Additionally, the concentration of chlorophyll a, b, and total and carotenoid increased with the application of TiO_2_ + AMF. In plants, N and P are known as structural elements in the formation of protein and chlorophylls [46]. It can be concluded that the AMF and plant roots’ symbiotic relationships provide the necessary nutrients for the biosynthesis of chlorophyll and carotenoid molecules; thereby, the concentration of the photosynthetic pigments increased. Moreover, the application of TiO_2_ acts as an anti-stress agent and induces an oxidation-reduction effect. In stressful conditions, the application of TiO_2_ protects the chloroplast through activating antioxidant enzymes such as superoxide dismutase, peroxidase, and catalase [20]. Chaudhary and Singh [18] noted that TiO_2_ nanoparticles could stabilize the integrality of chloroplast membrane and protect the chloroplasts against stressful conditions. Similarly, Morteza et al. [47] noted that the application of TiO_2_ enhanced maize productivity by increasing chlorophyll (a and b), carotenoid, and anthocyanin contents.

The RWC of sage leaves significantly decreased in drought stress conditions due to a lower water uptake by plant roots and greater water transpiration. In these conditions, inoculation of AMF with sage seedlings significantly enhanced the RWC content. It seems that the symbiotic relationship of AMF and penetration of fungi to plant roots improve the root hydraulic conductivity and provide a low-resistance path for water movement [48]. Hazzoumi et al. [49] reported that the application of AMF in water stress conditions enhanced the RWC content of basil (*Ocimum gratissimum* L) leaves.

Water use efficiency (WUE) is defined as the ratio of carbon assimilated as total biomass or grain yield per unit of water used by the crop. The highest WUE was achieved in no-stress irrigation regimes (MAD_25_) fertilized with TiO_2_ + AMF. It can be concluded that integrative application of biofertilizers and nanoparticles increased the DMY of sage as a result of the enhancement of nutrient uptake, plant growth characteristics, and photosynthetic rate, leading to an increase in assimilated carbon (as dry matter yield per unit of water used). In addition, the higher net income with the application of TiO_2_ + AMF under moderate water stress (MAD_50_) could be explained by the higher EO productivity and the positive role of the mentioned fertilizers in increasing dry matter productivity under stressful conditions.

## 4. Materials and Methods

### 4.1. Study Area

The present study was conducted during two growing years (2019 and 2020) at the research farm of Maragheh University, Iran (longitude 46°16′ E, latitude 37°22′ N, altitude 1532 m) (Figure 7). Meteorological data including monthly average temperature and precipitation were collected from the Iranian Meteorological Organization as shown in Table 5. Prior to any field activity, the soil physico-chemical properties of the experimental area (at a depth of 0–30 cm) were analyzed and are shown in Table 6.

### 4.2. Treatments, Land Preparation, and Cultivation

The experimental design was constituted by a split plot arranged in a randomized complete block design with three replicates. The irrigation treatments of plots included 25, 50, and 75% maximum allowable depletion (MAD) percentage of the soil available water as non-stress (MAD_25_), moderate (MAD_50_), and severe (MAD_75_) water stress. Subplots were given by four fertilizer sources including a no-fertilizer condition (control), TiO_2_ nanoparticles, AMF inoculation, and the co-addition of AMF + TiO_2_ nanoparticles. The seedlings of sage were sown on 8 and 12 April in 2019 and 2020, respectively, in rows with a spacing of 40 cm. Additionally, the on-row spacing of sage was set to be 30 cm. The rows were 3 m long. For TiO_2_ nanoparticle synthesis, TiO(OH)_2_ was produced by hydrolyzing and stirring titanium isopropoxide in an ice-cold (0 °C) condition. Then, titanyl nitrate [TiO(NO_3_)_2_] solution was obtained through dissolving TiO(OH)_2_ in nitric acid. Finally, titanyl nitrate and urea solution with a molar ratio of 1:1 was kept in a beaker (250 mL) and put into a muffle furnace maintained at 400 °C, and solid products were collected within 2 h [20]. TEM analysis was conducted at the Drug Applied Research Center, Tabriz University of Medical Sciences, Iran using Zeiss EM-90 operating at 80 kV tension (Figure 8a). Furthermore, particle size distribution was determined using dynamic light scattering (DLS) sizes by Zeta sizer Nano series (Nano ZS) (Malvern, ZEN3600). The TiO_2_ average particle size ranged between 60–70 nm (Figure 8b) and zeta potential was −11.1. Finally, the TiO_2_ was mixed with distilled water (100 mg L^−1^) and stirred at 35 °C for 2 h. After that, the obtained solution was sonicated via probe, ultrasonicated for 1 h to see the stable solution, and sprayed to sage seedlings in the pre-flowering stage at a concentration of 100 mg L^−1^ [20]. For AMF treatment, 80 g of soil, including spores of *Funneliformis mosseae* (obtained from Zist Fanavar Sabz Company, Iran), was added into planting holes.

### 4.3. Implementation of Irrigation Regimes

Firstly, for determining the soil water content (0–30 cm depth), a TDR probe (Model German, FM-Trime) was used, and after that, the irrigation depth of each treatment was calculated based on the following equations [27]:Soil available water (SAW) = (θ_fc_ − θ_pwp_) × soil layer depth × 100(1)
I_d_ = SAW × p(2)
I_n_= [I_d_ × 100]/E_a_(3)
where θ_fc_ is soil field capacity (27.1%), θ_pwp_ is soil permanent wilting point (13.7%), I_d_ is irrigation depth (cm), p is fraction of soil available water (25, 50 and 75%), E_a_ is irrigation efficiency (assumed 65%) [39], and I_n_ is gross depth of irrigation (cm). Based on the average of two years, the final irrigation volume of MAD_25_, MAD_50_, and MAD_75_ was 1.16, 0.87, and 0.58 m^3^ (= 1160, 870 and 580 L), respectively.

### 4.4. Measurements

#### 4.4.1. Yield, Yield Compound, and WUE

Before harvesting, morphological traits including plant height and canopy diameter were measured randomly in 10 seedlings in each plot. At the full flowering stage, the sage aerial parts were harvested randomly from a 1.8 m^2^ area on 20 August 2019 and 25 August 2020. The dry matter yield (DMY) was recorded after drying the harvested aerial parts at room temperature for two weeks. In addition, the water use efficiency index was calculated by dividing DMY by the volume of water consumption [50].

#### 4.4.2. Essential Oil and GC-MS Analysis

The EO percentage of sage was obtained using a Clevenger-type apparatus. First, 40 g of a ground sample was mixed with 300 mL of water and boiled for 3 h to distill the essential oil. The EO percentage was calculated based on the following equation [27]:EO content (*w*/*w*%) = [Extracted EO (g)/40 g] × 100(4)

In addition, the EO yield was calculated by multiplying the DMY with EO percentage and reported as g m^−2^. The sage EO compounds were analyzed using GC-MS and GC-FID instruments. In order to identify the sage essential oil components, an Agilent 7990 B gas chromatograph coupled to a 5988A mass spectrometer with an HP-5MS column (5% phenylmethyl polysiloxane, length 30 m, 0.25 mm internal diameter, and 0.25 μm film thickness) was utilized. The column temperature was initially set at 60 °C for 5 min, then was gradually increased to 240 °C at 3 °C min^−1^ and kept at 240 °C for 20 min. The carrier gas was helium (flow rate: 1 mL min^−1^). Mass spectra function parameters including electron impact, ionization temperature, and mass absorption range were set to 70 eV, 220 °C, and 40–400 *m z*^−1^, respectively. The injector was set in the split mode (split ratio 1:30). The injector and detector temperatures were maintained at 230 and 240 °C, respectively [51]. In order to calculate the peak retention index, a mixture of aliphatic hydrocarbons (C_8_-C_40_) was injected into the instrument under the same analytical conditions as the essential oil. The software used was chemstation. Identification of essential oil components was carried out using linear retention indices and mass spectra overlapping with commercial libraries [52]. Quantification was carried out using an Agilent 7990B gas chromatography device (USA) with a flame ionization detector (FID) and VF-5MS column (5% phenyl methylpolysiloxane, 30 m l., 0.25 mm i.d., 0.50 µm f.t.). The oven temperature was the same as that characterized for GC-MS analyses. EO samples were diluted (1:100) in hexane and 1 μL was injected. To determine the percentage of essential oil components, the peak area normalization method was used [28].

#### 4.4.3. AMF Root Colonization

To determine mycorrhizal root colonization, randomly collected root samples of sage were first rinsed to eliminate the soil attached to the roots. The root fragment was scavenged by heating for 15 min in 10% KOH. After three times of washing the root fragment with distilled water, the samples were mixed with 2% hydrochloric acid (HCL) for 20 min and stained by heating for 10 min in 0.05% trypan blue solution [53,54]. After staining, the root colonization percentage was measured based on the method of Giovannetti and Mosse [55]. Briefly, root samples were cut into 1 cm pieces, randomly were placed on a Petri dish, and evaluated by microscope (40 magnification). A grid line plate (dimensions 1 × 1 cm) was placed at the bottom of the Petri dish, and we counted the infected and non-infected roots that formed the intersection with vertical and horizontal lines of the grid line plate. Finally, the AMF colonization was calculated according to the following equation:Root colonization (%) = (number of infected root pieces /total root pieces) × 100(5)

#### 4.4.4. Nutrient Concentration

The nutrient concentration of sage aerial parts including N, P, and K was calculated based on the Kjeldahl method, flame photometry [56], and yellow method (using a spectrophotometer at 470 nm) [57], respectively.

#### 4.4.5. Chlorophyll and Carotenoid Content Measurements

To determine the chlorophyll (a, b, and total) and carotenoid content, 0.5 g of ground sage leaves (in liquid N) was mixed with 10 mL of 80% acetone, and the absorbance was read at 646.8, 663.2, and 470 nm using a spectrophotometer (UV-1800, Shimadzu, Japan). The concentration of photosynthetic pigments was measured based on the following equations [58]:Chl a = (12.25 A_663.2_) − (2.79 A_646.8_)(6)
Chl b = (21.5 A_646.8_) − (5.1 A_663.2_)(7)
Chl a + b = (7.15 A_663.2_) + (18.71 A_646.8_)(8)
Car = [1000 A_470_ − 1.82 C_a_ − 85.02 C_b_]/198(9)

#### 4.4.6. Relative Water Content (RWC)

The relative water content (RWC) of sage leaves was measured according to the procedures reported by Levitt [59]:(10)$$\mathrm{RWC}\;(\%)=\frac{\mathrm{Fw}-\mathrm{Dw}}{\mathrm{Tw}-\mathrm{Dw}}\;\times \;100$$

In this equation, F_w_, T_w_, and D_w_ are the fresh weight of leaves, turgid weight (by floating the fresh leaves in double distilled water at 4 °C for 5 h), and dry weight (by drying the leaves in an oven at 70 °C for 48 h), respectively.

### 4.5. Statistical Analysis

The statistical analyses, including an analysis of the combined variance (ANOVA) and the comparison of data means, were performed by the LSD test at the *p* < 0.05 level using SAS v9.3 (SAS Institute, Cary, NC, USA) and SPSS v25 software (IBM Corp., Armonk, NY, USA), and the graphs were drawn in Excel. The TiO_2_ nanoparticles and AMF application were considered as fixed effects, while block, year, and their interactions were random effects. Additionally, R software v3.2.4 (R Foundation for Statistical Computing, Vienna, Austria) was used to analyze Pearson’s correlation between the DMY, EO content, and yield, WUE, RWC, chlorophyll a, chlorophyll b, chlorophyll total, carotenoids, N, P, and K along with *cis*-thujone, camphor, and 1,8-cineole percentages.

## 5. Conclusions

The results of study showed that drought significantly decreased sage DMY, nutrient acquisition, photosynthetic potential, and WUE, which is expected in semi-arid and arid regions. In this study, the application of a biofertilizer and nanoparticles (AMF + TiO_2_) increased sage productivity in drought stress conditions. As hypothesized in this study, moderate drought stress (MAD_50_) could increase EO content and EO yield along with the main EO constituent percentages when sage was fertilized with TiO_2_ + AMF. Furthermore, the co-addition of TiO_2_ + AMF increased WUE, which is important in water-limited regions. Finally, the increase in the EO productivity with the application of TiO_2_ + AMF led to an increase in net income in drought stress conditions. Overall, the results of study demonstrated that the integrative application of TiO_2_ + AMF could be recommended as a sustainable management method for improving EO quantity and quality of sage in drought stress conditions. Future research should be focused on the effects of different nanoparticles as well as different AMF species for alleviating drought stress impacts and improving EO quantity and quality of medicinal and aromatic plants.

## Figures and Tables

**Figure 1 plants-11-01659-f001:**
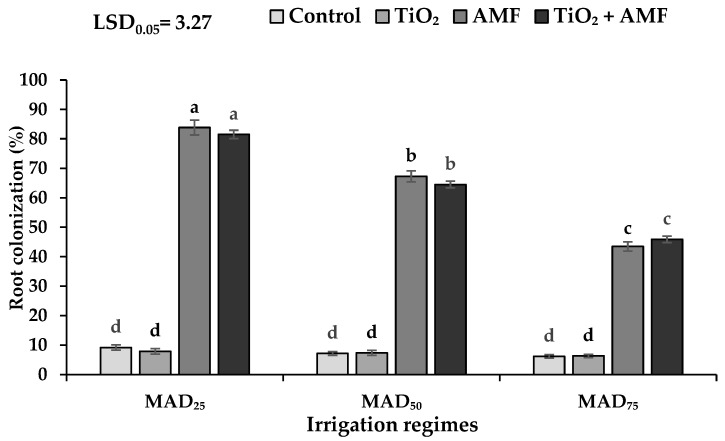
The root colonization of sage as influenced by different irrigation regimes and fertilizer sources. MAD_25_, MAD_50_, and MAD_75_ indicating non-stressed, moderate, and severe drought stress, respectively. AMF: arbuscular mycorrhizal fungi; TiO_2_: titanium dioxide nanoparticle. Different letters indicate significant differences at the 5% level according to least significant difference (LSD) test.

**Figure 2 plants-11-01659-f002:**
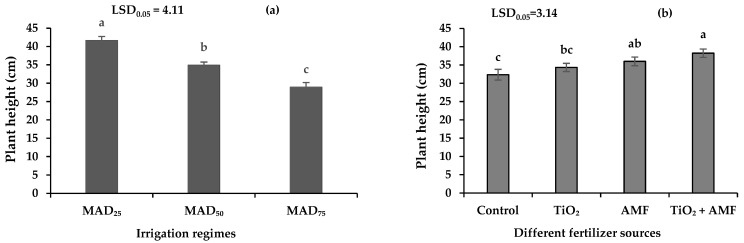
The plant height of sage in different irrigation regimes (**a**) and fertilizer sources (**b**). MAD_25_, MAD_50_, and MAD_75_ indicating non-stressed, moderate, and severe drought stress, respectively. AMF: arbuscular mycorrhizal fungi; TiO_2_: titanium dioxide nanoparticle. Different letters indicate significant differences at the 5% level according to LSD test.

**Figure 3 plants-11-01659-f003:**
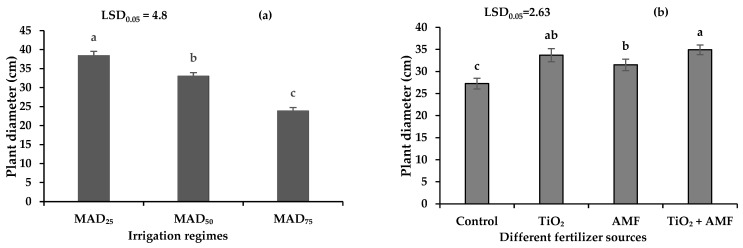
The plant diameter of sage in different irrigation regimes (**a**) and fertilizer sources (**b**). MAD_25_, MAD_50_, and MAD_75_ indicating non-stressed, moderate, and severe drought stress, respectively. AMF: arbuscular mycorrhizal fungi; TiO_2_: titanium dioxide nanoparticle. Different letters indicate significant differences at the 5% level according to LSD test.

**Figure 4 plants-11-01659-f004:**
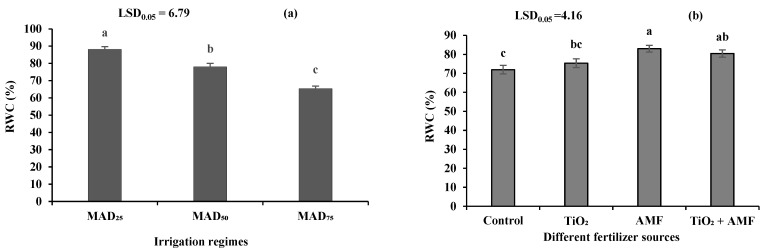
The relative water content (RWC) of sage in different irrigation regimes (**a**) and fertilizer sources (**b**). MAD_25_, MAD_50_, and MAD_75_ indicating non-stressed, moderate, and severe drought stress, respectively. AMF: arbuscular mycorrhizal fungi; TiO_2_: titanium dioxide nanoparticle. Different letters indicate significant differences at the 5% level according to LSD test.

**Figure 5 plants-11-01659-f005:**
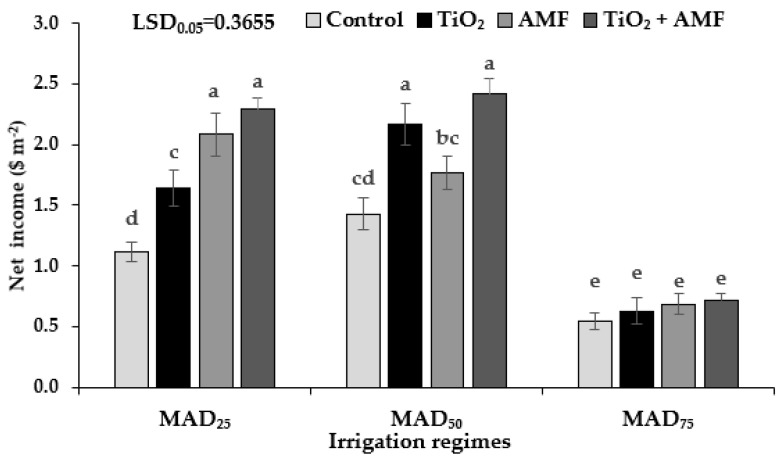
The net income of sage in different irrigation regimes and fertilizer sources. MAD_25_, MAD_50_, and MAD_75_ indicating non-stressed, moderate, and severe drought stress, respectively. AMF: arbuscular mycorrhizal fungi; TiO_2_: titanium dioxide nanoparticle. Different letters indicate significant differences at the 5% level according to LSD test.

**Figure 6 plants-11-01659-f006:**
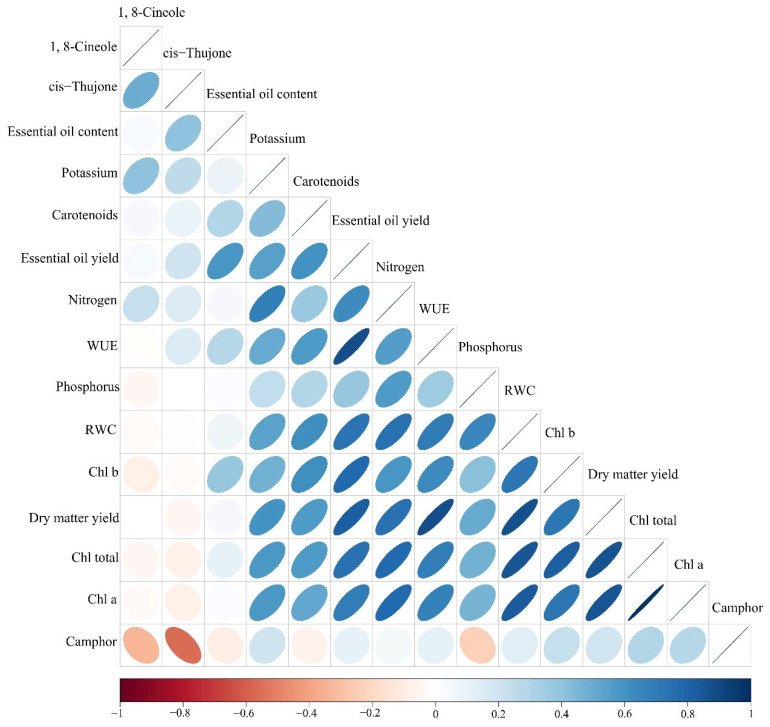
Pearson’s correlation matrix for studied traits of sage. Color ellipses illustrate statistically significant levels. Positive and negative correlations are shown in blue and red, respectively. The color legend on the bottom-hand side of corrplot represent the intensities of Pearson correlation coefficients.

**Figure 7 plants-11-01659-f007:**
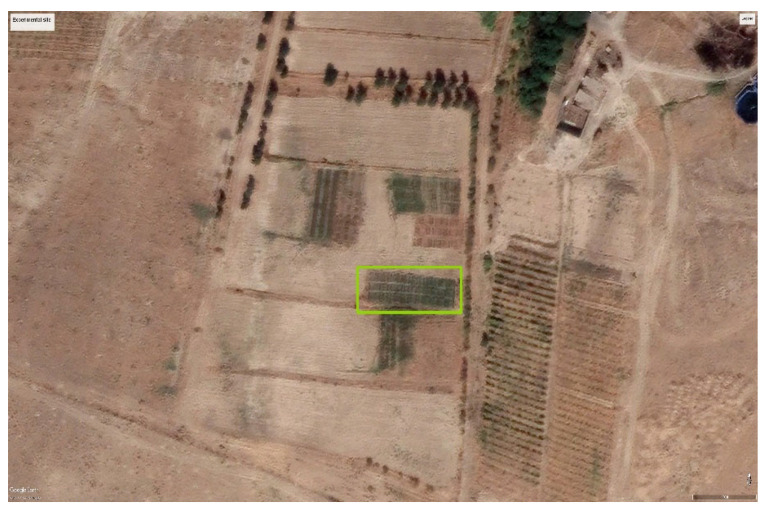
Aerial view of the experimental area (longitude 46°16′ E, latitude 37°22′ N, altitude 1532 m). The experimental area is shown by green mark.

**Figure 8 plants-11-01659-f008:**
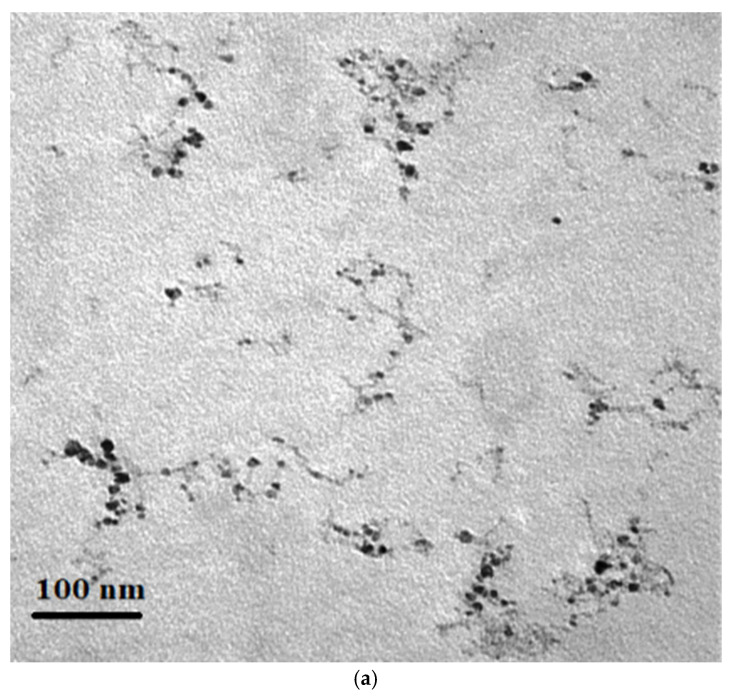
TEM image (**a**) and dynamic light scattering (DLS) of TiO_2_ nanoparticles (**b**).

**Table 1 plants-11-01659-t001:** The concentration nitrogen (N), phosphorus (P), and potassium (K) of sage in different irrigation regimes and fertilizer sources.

Treatments		Nitrogen (g kg^−1^)	Phosphorus (g kg^−1^)	Potassium (g kg^−1^)
MAD_25_	Control	23.17 ^g^	1.74 ^d,e^	22.55 ^f^
	TiO_2_	28.40 ^b^	1.97 ^b^	25.22 ^d^
	AMF	27.91 ^c^	2.02 ^b^	27.24 ^b^
	TiO_2_ + AMF	29.88 ^a^	2.20 ^a^	29.73 ^a^
MAD_50_	Control	23.57 ^f^	1.58 ^h^	22.09 ^g,h^
	TiO_2_	24.16 ^e^	1.69 ^e,f^	22.36 ^f,g^
	AMF	23.97 ^e^	1.83 ^c^	26.86 ^c^
	TiO_2_ + AMF	24.95 ^d^	1.79 ^c,d^	23.92 ^e^
MAD_75_	Control	21.01 ^i^	1.47 ^i^	19.60 ^j^
	TiO_2_	21.80 ^h^	1.59 ^g,h^	20.39 ^i^
	AMF	21.80 ^h^	1.65 ^f,g^	20.66 ^i^
	TiO_2_ + AMF	21.71 ^h^	1.57 ^h^	21.90 ^h^
LSD_0.05_		0.3016	0.0614	0.3285
Significance		Significance levels		
Year (Y)		ns	ns	ns
Irrigation regimes (W)		**	*	**
Y × W		ns	ns	ns
Fertilization (F)		**	**	**
W × F		**	**	**
Y × F		ns	ns	ns
W × F × Y		ns	ns	ns

MAD_25_, MAD_50_, and MAD_75_ indicating non-stressed, moderate, and severe drought stress, respectively. AMF: arbuscular mycorrhizal fungi; TiO_2_: titanium dioxide nanoparticle. Ns, *, and ** indicate no significant difference, significant at 5% probability level, and significant at 1% probability level, respectively. Different letters indicate significant differences at the 5% level according to LSD test.

**Table 2 plants-11-01659-t002:** The dry matter yield, essential oil content, essential oil yield, and water use efficiency of sage in different irrigation regimes and fertilizer sources.

Treatments		Dry Matter Yield (g m^−2^)	Essential Oil Content (%)	Essential Oil Yield (g m^−2^)	Water Use Efficiency (g m^−2^)
MAD_25_	Control	185.12 ^b,c^	0.615 ^e^	1.14 ^f,g^	159.78 ^c,d,e^
	TiO_2_	193.07 ^b,c^	0.887 ^d^	1.72 ^d,e^	166.65 ^b,c,d,e^
	AMF	221.98 ^b^	0.963 ^cd^	2.14 ^b,c^	191.60 ^b,c^
	TiO_2_ + AMF	262.85 ^a^	0.915 ^d^	2.41 ^a,b^	226.83 ^a^
MAD_50_	Control	134.62 ^d^	1.067 ^c^	1.45 ^e,f^	154.98 ^d,e,f^
	TiO_2_	158.75 ^cd^	1.410 ^a^	2.24 ^a,b^	182.78 ^b,c,d^
	AMF	143.63 ^d^	1.268 ^b^	1.82 ^c,d^	165.38 ^b,c,d,e^
	TiO_2_ + AMF	169.83 ^c,d^	1.483 ^a^	2.52 ^a^	195.55 ^a,b^
MAD_75_	Control	62.75 ^e^	0.885 ^d^	0.56 ^h^	108.35 ^g^
	TiO_2_	71.93 ^e^	0.948 ^d^	0.69 ^h^	124.18 ^f,g^
	AMF	78.72 ^e^	0.918 ^d^	0.73 ^h^	135.87 ^e,f,g^
	TiO_2_ + AMF	85.50 ^e^	0.948 ^d^	0.81 ^g,h^	147.57 ^e,f^
LSD_0.05_		38.982	0.117	0.3643	33.94
Source of variation		Significance levels			
Year (Y)		ns	ns	ns	ns
Irrigation regimes (W)		*	*	**	**
Y × W		ns	*	ns	ns
Fertilization (F)		**	**	*	*
W × F		*	*	*	*
Y × F		ns	ns	ns	ns
W × F × Y		ns	ns	ns	ns

MAD_25_, MAD_50_, and MAD_75_ indicating non-stressed, moderate, and severe drought stress, respectively. AMF: arbuscular mycorrhizal fungi; TiO_2_: titanium dioxide nanoparticle. Ns, *, and ** indicate no significant difference, significant at 5% probability level, and significant at 1% probability level, respectively. Different letters indicate significant differences at the 5% level according to LSD’s test.

**Table 3 plants-11-01659-t003:** The essential oil constituents of sage in different irrigation regimes and fertilizer sources (average of two years).

No	RI ^a^	LIT.RI ^b^	Components	Treatments ^c^											
				MAD_25_ Control	MAD_25_ TiO_2_	MAD_25_ AMF	MAD_25_ TiO_2_ + AMF	MAD_50_ Control	MAD_50_ TiO_2_	MAD_50_ AMF	MAD_50_ TiO_2_ + AMF	MAD_75_ Control	MAD_75_ TiO_2_	MAD_75_ AMF	MAD_75_ TiO_2_ + AMF
1	843	847	*cis*-Salvene	0.36	0.47	0.39	0.46	0.48	0.44	0.56	0.52	0.54	0.44	0.50	0.44
2	853	858	*trans*-Salvene	0.10	0.09	0.11	0.09	0.12	0.12	0.13	0.11	0.11	0.17	0.00	0.11
3	919	921	Tricyclene	0.15	0.14	0.14	0.12	0.13	0.12	0.15	0.13	0.14	0.12	0.14	0.13
4	924	924	*α*-Thujene	0.15	0.16	0.18	0.21	0.20	0.20	0.19	0.21	0.18	0.17	0.20	0.14
5	930	932	*α*-Pinene	2.51	3.34	3.33	3.13	3.65	3.46	3.49	3.88	3.56	2.69	3.67	3.34
6	944	946	Camphene	3.11	3.44	3.32	3.58	3.71	3.64	4.00	4.35	3.87	3.15	4.17	3.40
7	972	974	*β*-Pinene	1.26	1.29	1.47	1.28	1.39	1.44	1.49	1.35	1.51	1.29	1.49	1.24
8	989	988	Myrcene	0.94	0.96	1.07	0.94	0.98	0.95	1.01	1.03	1.02	0.97	0.91	0.93
9	999	1000	*n*-Decane	0.13	0.12	0.14	0.14	0.12	0.12	0.12	0.12	0.12	0.13	0.12	0.13
10	1014	1014	*α*-Terpinene	0.31	0.39	0.40	0.36	0.38	0.38	0.40	0.42	0.37	0.40	0.38	0.40
11	1021	1022	*ρ*-Cymene	1.44	1.41	1.50	1.35	1.55	1.35	1.52	1.40	1.59	1.37	1.61	1.44
12	1028	1026	1,8-Cineole	9.17	10.49	10.37	9.89	9.25	9.87	10.57	9.56	9.98	10.71	9.91	10.30
13	1055	1054	*γ*-Terpinene	0.36	0.34	0.36	0.33	0.34	0.31	0.34	0.37	0.40	0.38	0.39	0.33
14	1063	1065	*cis*-Sabinene hydrate	0.24	0.24	0.23	0.18	0.19	0.21	0.21	0.24	0.20	0.21	0.26	0.17
15	1085	1086	Terpinolene	0.20	0.16	0.16	0.15	0.21	0.16	0.17	0.17	0.20	0.17	0.19	0.14
16	1096	1098	*trans*-Sabinene hydrate	0.66	0.44	0.41	0.45	0.45	0.41	0.38	0.23	0.41	0.50	0.41	0.42
17	1105	1101	*cis*-Thujone	30.73	33.63	34.70	33.89	31.41	33.55	33.37	35.84	33.01	33.99	32.32	32.08
18	1113	1112	*trans*-Thujone	4.59	4.97	4.94	5.30	5.04	5.53	4.91	5.68	4.97	5.52	4.87	5.07
19	1140	1141	Camphor	17.48	17.19	14.58	16.88	17.02	16.02	17.95	15.54	15.77	15.07	18.36	18.61
20	1146	1147	*neo*-*iso*-3-Thujanol	0.84	0.51	0.46	0.57	0.57	0.52	0.50	0.47	0.55	0.59	0.31	0.53
21	1155	1158	*trans*-Pinocamphone	0.12	0.12	0.05	0.11	0.13	0.06	0.12	0.06	0.10	0.15	0.05	0.12
22	1161	1165	Borneol	3.64	2.16	2.06	3.23	2.88	2.67	2.44	1.94	3.12	2.83	3.17	2.56
23	1169	1172	*cis*-Pinocamphone	0.60	0.32	0.35	0.36	0.34	0.37	0.36	0.34	0.35	0.36	0.38	0.34
24	1173	1174	Terpinen-4-ol	0.06	0.10	0.10	0.06	0.06	0.06	0.07	0.06	0.06	0.07	0.03	0.09
25	1186	1186	*α*-Terpineol	0.15	0.19	0.15	0.14	0.14	0.14	0.17	0.13	0.19	0.21	0.08	0.16
26	1282	1284	Bornyl acetate	2.27	1.57	1.84	1.80	1.94	1.69	1.54	1.42	2.21	2.25	2.50	1.41
27	1290	1289	*trans*-Sabinyl acetate	0.00	0.12	0.08	0.13	0.08	0.06	0.07	0.11	0.08	0.09	0.00	0.10
28	1413	1417	(*E*)-Caryophyllene	2.16	2.32	2.30	2.13	2.41	2.05	1.94	2.42	1.92	2.07	1.95	2.32
29	1447	1452	*α*-Humulene	2.64	2.84	2.81	2.63	3.04	2.65	2.49	3.35	2.61	2.77	2.52	3.22
30	1576	1582	Caryophyllene oxide	0.38	0.36	0.35	0.36	0.42	0.43	0.33	0.40	0.39	0.40	0.36	0.30
31	1585	1592	Viridiflorol	4.19	4.10	4.20	3.37	4.11	4.37	3.16	3.76	3.69	3.87	3.73	4.02
32	1602	1608	Humulene epoxide II	0.61	0.64	0.75	0.75	0.79	0.77	0.60	0.68	0.75	0.74	0.64	0.68
33	2051	2056	Manool	2.82	2.18	2.86	2.65	3.04	3.01	2.16	2.42	2.78	2.53	2.68	2.34
Total identified (%)				94.37	96.8	96.16	97.02	96.57	97.13	96.91	98.71	96.75	96.38	98.30	97.01
Grouped compounds (%)															
Monoterpene hydrocarbons				10.80	11.99	12.30	11.77	12.85	12.34	13.09	13.67	13.16	11.05	13.53	11.79
Oxygenated monoterpenes				68.04	70.12	68.17	70.88	67.29	69.20	70.84	69.85	68.51	70.00	69.89	70.28
Sesquiterpene hydrocarbons				4.80	5.16	5.11	4.76	5.45	4.70	4.43	5.77	4.53	4.84	4.47	5.54
Oxygenated sesquiterpenes				5.18	5.10	5.30	4.48	5.32	5.57	4.09	4.84	4.83	5.01	4.73	5.00
Oxygenated diterpenes				2.82	2.18	2.86	2.65	3.04	3.01	2.16	2.42	2.78	2.53	2.68	2.34
Others				2.73	2.25	2.42	2.48	2.62	2.31	2.30	2.16	2.94	2.95	3.00	2.06

^a^ RI, linear retention indices on DB-5 MS column, experimentally determined using homologue series of n-alkanes. ^b^ Relative retention indices taken from Adams. ^c^ MAD_25_, MAD_50_, and MAD_75_ indicating non-stressed, moderate, and severe drought stress, respectively. Control: no fertilizer, AMF: arbuscular mycorrhizal fungi, TiO_2_: titanium dioxide nanoparticle.

**Table 4 plants-11-01659-t004:** Chlorophyll a, b, and total and carotenoid of sage in different irrigation regimes and fertilizer sources.

Treatments		Chlorophyll a (mg g^−1^ Fresh Weight)	Chlorophyll b (mg g^−1^ Fresh Weight)	Chlorophyll Total (mg g^−1^ Fresh Weight)	Carotenoid (mg g^−1^ Fresh Weight)
MAD_25_	Control	3.89 ^d^	0.92 ^e^	4.81 ^d^	1.54 ^d^
	TiO_2_	4.42 ^c^	1.08 ^c^	5.21 ^c^	1.35 ^e^
	AMF	5.00 ^b^	0.99 ^d^	5.99 ^b^	1.49 ^d^
	TiO_2_ + AMF	5.56 ^a^	1.30 ^a^	6.85 ^a^	1.78 ^a,b^
MAD_50_	Control	2.95 ^f^	0.95 ^d,e^	3.90 ^f^	1.72 ^b^
	TiO_2_	3.35 ^e^	0.79 ^f^	4.43 ^e^	1.70 ^b,c^
	AMF	2.52 ^g^	0.98 ^d,e^	3.50 ^g^	1.90 ^a^
	TiO_2_ + AMF	3.56 ^d,e^	1.22 ^b^	4.78 ^d^	1.75 ^b^
MAD_75_	Control	1.09 ^h^	0.22 ^i^	1.31 ^j^	0.44 ^f^
	TiO_2_	1.36 ^h^	0.47 ^g^	1.83 ^i^	0.46 ^f^
	AMF	1.18 ^h^	0.22 ^i^	1.40 ^j^	0.50 ^f^
	TiO_2_ + AMF	2.40 ^g^	0.30 ^h^	2.70 ^h^	1.58 ^c,d^
LSD_0.05_		0.33	0.056	0.31	0.131
Source of variation		Significance levels			
Year (Y)		ns	ns	ns	ns
Irrigation regimes (W)		**	**	**	**
Y × W		ns	ns	ns	ns
Fertilization (F)		**	**	**	**
W × F		**	**	**	**
Y × F		ns	ns	ns	ns
W × F × Y		ns	ns	ns	ns

MAD_25_, MAD_50_, and MAD_75_ indicating non-stressed, moderate., and severe drought stress, respectively. AMF: arbuscular mycorrhizal fungi; TiO_2_: titanium dioxide nanoparticle. Ns and ** indicated no significant difference, significant at 5% probability level, and significant at 1% probability level, respectively. Different letters indicate significant differences at the 5% level according to LSD test.

**Table 5 plants-11-01659-t005:** Monthly average temperature and total monthly precipitation in 2019 and 2020 growing seasons and long-term averages in the experimental area.

Year	April	May	June	July	August	September
Monthly average temperature (°C)						
2019	10.4	18.5	25.7	27.6	27.8	22.1
2020	11.8	19.1	24.2	28.0	25.1	23.8
2-year mean	11.1	18.8	25.0	27.8	26.5	23.0
10-year mean	12.6	18.2	24.1	28.1	27.5	22.7
Total monthly precipitation (mm)						
2019	51.3	37.8	4.2	0.0	0.0	0.0
2020	63.3	12.0	2.6	0.1	1.2	0.0
2-year mean	57.3	24.9	3.4	0.1	0.6	0.0
10-year mean	44.8	20.6	1.7	0.5	0.4	2.0

**Table 6 plants-11-01659-t006:** Physico-chemical properties of field soil on average over the 2 years.

Soil Texture	Sand (%)	Silt (%)	Clay (%)	Organic Matter (g kg^−1^)	EC (ds.m^−1^)	pH	Field Capacity (%)	Permanent Wilting Point (%)	Exchangeable Potassium (mg kg^−1^)	Cation Exchange Capacity (Cmolc kg^−1^)	Available Phosphorus (mg kg^−1^)	Total Nitrogen (g kg^−1^)
Sandy clay loam	56.3	16.3	27.4	8.1	1.17	7.73	27.1	13.7	563.85	26.5	9.7	0.87

## Data Availability

The datasets generated and analyzed during the current study are available from the corresponding author upon reasonable request.

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
