# Peer review of "Co-Application of TiO2 Nanoparticles and Arbuscular Mycorrhizal Fungi Improves Essential Oil Quantity and Quality of Sage (Salvia officinalis L.) in Drought Stress Conditions"

_plants, 2022, doi:10.3390/plants11131659_

Round 1

Author Response

# Reviewer 1

This study is relevant, the use of nanofertilizers and fungi to increase plant resistance to stress has recently been used in modern crop production. The possible effect of their joint composition is of interest. I think the work is relevant, interesting and done at a good methodological level. During the analysis of the manuscript, there were points that need to be corrected. This will make it easier and better to understand the work.

Reply: Thank you very much for your valuable comments to increase the quality of the article and the corrections were made based on the reviewer comments.

- I am glad that the authors put forward the hypothesis of line 98-100. In the "Conclusion" section, it is worth emphasizing that your hypothesis is correct.

Reply: Thanks. The corrections were made in conclusion section. We added: ‘Based on our hypothesis, moderate drought stress (MAD50) could increase EO content and EO yield along with the main EO constituent percentages when sage was fertilized with TiO2 + AMF. Also, Co-addition of TiO2 + AMF increased WUE which is important in water limited regions.’

- in the "Results" section, based on the data in Figure 1, we can conclude that the combined use of titanium oxide and the fungus still increases the colonization of the fungus under MAD75 conditions. At the same time, the level of colonization, such as under the action of only AMF. The text says «The rate of AMF colonization in MAD50 and MAD75 decreased by 19.8 and 44.14% in comparison with no stress conditions (MAD25), respectively» Perhaps it is worth constructing the phrase differently here, since you still observe an increase in the percentage of colonization at MAD75.

Reply: We revised this sentence based on the reviewer comment. We revised: ‘Although the AMF colonization increased in drought stress conditions (MAD50 and MAD75) in AMF and TiO2 + AMF treatments (Fig. 1).’

- in Fig. 2, 3, 4 (b) it is necessary to indicate at what MAD you considered the studied parameters in these figures, it is not entirely clear.

Reply: Based the analysis variance results (ANOVA), the plant height, canopy diameter and relative water content (RWC) of sage was affected by single effects of irrigation regimes and fertilizer but not by their interaction. Therefore, in the mentioned traits, the means of different irrigation regimes and fertilizer sources were examined differently. The figures of 2A, 3A and 4A was reported the effects of different irrigation regimes on the plant height, canopy diameter and RWC content and also the figures of 2B, 3B and 4B was reported the effects of different fertilizer sources on the plant height, canopy diameter and RWC content of sage seedlings. In addition, the ANOVA results was reported briefly in the first lines of each measured traits.

- In the «Discussion» section, you discussed what each of the added components does individually. Give a small conclusion which of the components contributes to which indicator. Everything is written correctly, I want specifics regarding exactly what they do together. Since they have a very good effect at MAD 75, for example, on photosynthetic pigments and indicators of water use and other indicators.

 Reply: Thanks for valuable comment. The aim of this study was improving nutrient accessibility in drought stress conditions which lead to increasing plant performance, essential oil productivity as well as essential oil quality and also physiological properties of sage seedlings under stressful conditions. Therefore, we revised some sections based on the reviewer comments:

‘Therefore, the higher productivity of sage seedlings with integrative application of TiO2 + AMF could be explained by the role of the mentioned fertilizers in increasing the nutrients availability which will lead to optimal plant growth performance.’

‘It seems that the integrative application of TiO2 + AMF improves the nutrients accessibility as a result of stronger seedlings and higher metabolic efficiency (e.g., nutrient transport) that have an important role in the production of carbohydrates, and development of the glandular trichomes, EO channels and secretory ducts.’

‘It can be concluded that the AMF/plant roots symbiotic relationships provide the necessary nutrients for the biosynthesis of chlorophyll and carotenoid molecules, thereby the concentration of the photosynthetic pigments increased.’

- Raises the question of how much it costs to use nanofertilizers in general? How much it reduces the cost or makes the final product more expensive. And to what extent such treatments can be applied to cereals, for example, wheat

Reply: In the last century, the average temperature of the earth has increased by 0.7 °C leading to transformation of temperate climates into semi-arid and arid climates. More than 88% of farmlands in Iran are in arid and semi-arid regions. Drought negatively affects nutrient absorption, photosynthetic capacity, and water use efficiency (WUE) of plants which leads to reduction in plant productivity and quality. In addition, water scarcity reduces the uptake rate of nutrients in the soil by reducing root absorption capacity and decreasing rate of nutrient diffusion from the soil to the root uptake. For example, 40–70%, 80–90%, and 50–90% of nitrogen (N), phosphorus (P), and potassium (K) fertilizers are lost and/or fixed in soils, resulting in economic losses. In these conditions, the excessive using of chemical fertilizer enhanced production costs. Therefore, the use of alternative fertilizers as an effort to increase the absorption rate of nutrients, especially in drought stress conditions is necessary. Nanotechnology is a modern strategy that has shown to increase the tolerance of agricultural crops towards drought stress. Nanotechnology bridges the gap in the loss of nutrients and fortification of plants, as applied to agriculture. Using recent nanotechnologies, urban agriculture could make an enormous contribution to food safety and to wellness nutrition.

The main objective of this study is increasing the nutrient use efficiency of sage plants under drought stress conditions through the replacement of chemical fertilizers with low-cost and eco-friendly fertilizers such as nanofertilizers and inoculating plants with biofertilizers such as AMF. In this study, we assessed the net income in different irrigation regimes and fertilizers sources. Based on the obtained results, AMF inoculation or co-addition of TiO2 + AMF are most economical decisions for sage growers. Also, in moderate drought condition (MAD50), fertilization with TiO2 or co-addition of TiO2 + AMF are effective strategies to increase sage growers’ income. However, the production costs based on the using of chemical fertilizers is not determined in this study. However, in subsequent experiments, the productivity yield and also production cost of agricultural plants with application of chemical fertilizers, nano- and biofertilizers in drought stress conditions can be evaluated.

Reviewer 2 Report

Article “Co-Application of TiO2 Nanoparticles and Arbuscular Mycorrhizal Fungi

Improves Essential Oil Quantity and Quality of Sage (Salvia officinalis L.) in Drought Stress Conditions” Ostadi et al. is devoted to the effect of nanofertilizers based on titanium oxide and abuscular fungus on the physiological and biochemical parameters of the sage plant under different drought conditions.

This study is relevant, the use of nanofertilizers and fungi to increase plant resistance to stress has recently been used in modern crop production. The possible effect of their joint composition is of interest. I think the work is relevant, interesting and done at a good methodological level. During the analysis of the manuscript, there were points that need to be corrected. This will make it easier and better to understand the work.

*I am glad that the authors put forward the hypothesis of line 98-100. In the "Conclusion" section, it is worth emphasizing that your hypothesis is correct.

*in the "Results" section,

**Based on the data in Figure 1, we can conclude that the combined use of titanium oxide and the fungus still increases the colonization of the fungus under MAD75 conditions.

At the same time, the level of colonization, such as under the action of only AMF. The text says «The rate of AMF colonization in MAD50 and MAD75 decreased by 19.8 and 44.14% in comparison with no stress conditions (MAD25), respectively» Perhaps it is worth constructing the phrase differently here, since you still observe an increase in the percentage of colonization at MAD75.

*** in Fig. 2, 3, 4 (b) it is necessary to indicate at what MAD you considered the studied parameters in these figures, it is not entirely clear.

      In the «Discussion» section, you discussed what each of the added components does individually. Give a small conclusion which of the components contributes to which indicator. Everything is written correctly, I want specifics regarding exactly what they do together. Since they have a very good effect at MAD 75, for example, on photosynthetic pigments and indicators of water use and other indicators.

Raises the question of how much it costs to use nanofertilizers in general? How much it reduces the cost or makes the final product more expensive. And to what extent such treatments can be applied to cereals, for example, wheat

Author Response

# Reviewer 2

The general idea of the manuscript is of potential interest and the study was well conducted but the authors should clarify some important issues, as follows:

  1. In the Introduction, please explain why the authors used TIO2 for minimizing the effects of different stress conditions in crop production and its quality. Please add information about the previous effect of TiO2 on essential oil content, nutrient concentration, chlorophyll and carotenoid content in other plants. What are the mechanisms involved? How do you explain these increases under the action of TiO2?

Reply: We added some previous study reports about application of TiO2 nanoparticles enhance essential oil quantity and its quality under drought stress conditions in introduction section. WE added:

‘Among different nanoparticles, TiO2 nanoparticles have positive impacts on the plant growth such as increasing nutrient uptakes, improving chlorophyll content and promote light capture in chlorophylls (a and b), regulation of important enzymes activities such as glutamine synthase, glutamate dehydrogenase and also nitrate reductase, chemical fixation of dinitrogen (N2) in the air, electron transfer activities, improving carbon dioxide (CO2) assimilation and photosynthetic activities [15, 16]. In addition, TiO2, as an anti-stress agent, decrease the negative effects of environmental stress by increasing the antioxidant enzymes activity [17].’

‘Previous studies reported the positive effects of TiO2 nanoparticles on the improvement of plant performance, especially medicinal and aromatic plant, in stressful conditions. Ah-mad et al. [18] noted that application of TiO2 nanoparticles increased the EO quantity and yield by 39 and 105% in peppermint seedlings. Gohari et al. [17] noted that application of TiO2 nanoparticles improve EO quantity and quality of Dracocephalum moldavica L. in stressful conditions.’

‘Also, Khater et al. [21] report ed that the foliar application of TiO2 nanoparticles im-proved EO quality of coriander through increasing the main EO constituents such as linalool.’

‘Shabbir et al. [22] reported that the application of TiO2 nanoparticles enhanced dry matter yield of vetiver (Vetiveria zizanioides L. Nash) as well as physiological characteristics such as total chlorophyll content, net photosynthetic rate, intercellular CO2 concentration, car-bonic anhydrase and nitrate reductase activity. Also, the authors noted that the EO content and yield and also khusimol (main active constituent of EO) content increased by 23.6, 55.1 and 24.5% with application of TiO2 nanoparticles.’

  1. Please add in Material and Methods information about the characterization of TiO2 nanoparticles used (size, zeta potential, method of preparation).

Reply: The method of preparation, the Transmission electron microscopy (TEM) image, dynamic light scattering (DLS) and zeta potential of TiO2 nanoparticles was added in text.

We added: For TiO2 nanoparticles synthesis, TiO(OH)2 was produced by hydrolyzing and stirring of titanium isopropoxide in ice-cold (0 °C) condition. Then, titanyl nitrate [TiO(NO3)2] solu-tion was obtained through dissolving of TiO(OH)2 in nitric acid. Finally, titanyl nitrate and urea solution with molar ratio of 1:1 was kept in beaker (250 mL) beaker and put into a muffle furnace maintained at 400 °C and solid products were collected within 2h [20]. TEM analysis was done at Drug Applied Research Center, Tabriz University of Medical Sciences, Iran with using of Zeiss EM-90 operating at 80 kV tension (Figure 8a). Also, par-ticle size distribution was determined using dynamic light scattering (DLS) sizes by Zeta sizer Nano series (Nano ZS) (Malvern, ZEN3600). The TiO2 average particles size was ranges between 60-70 nm (Figure 8b) and zeta potential was -11.1. Finally, the TiO2 was mixed with distilled water (100 mg L -1) and stirred at 35 °C for 2h. After that, the obtained solution has been sonicated via probe ultrasonicated for 1h until to see the stable solution and sprayed to sage seedlings in the pre-flowering stage at concentration of 100 mg L-1.

  1. Table 1, for example, does not show that there are significant differences between the effect of TiO2 and TiO2 + AFM compared to AFM. In my opinion, the best results are obtained with AFM or AFM + TiO2 but also thanks to AFM. Please explain why. Which are the mechanisms involved in TiO2 action on relative water content of sage, on essential oil content, water used efficiency, chlorophyll and carotenoid content?

Reply: Thanks for valuable comment. The aim of this study was improving nutrient accessibility in drought stress conditions which lead to increasing plant performance, essential oil productivity as well as essential oil quality and also physiological properties of sage seedlings under stressful conditions. Two strategies to alleviate drought stress effects on plant production and quality are rep-resented by harnessing advances in nanotechnology such as application of TiO2 and inoculating plants with biofertilizers such as arbuscular mycorrhizae Fungi (AMF). Therefore, we revised some sections based on the reviewer comments:

‘Therefore, the higher productivity of sage seedlings with integrative application of TiO2 + AMF could be explained by the role of the mentioned fertilizers in increasing the nutrients availability which will lead to optimal plant growth performance.’

‘It seems that the integrative application of TiO2 + AMF improves the nutrients accessibility as a result of stronger seedlings and higher metabolic efficiency (e.g., nutrient transport) that have an important role in the production of carbohydrates, and development of the glandular trichomes, EO channels and secretory ducts.’

‘It can be concluded that the AMF/plant roots symbiotic relationships provide the necessary nutrients for the biosynthesis of chlorophyll and carotenoid molecules, thereby the concentration of the photosynthetic pigments increased.’

Also, we added the main reason of increasing chlorophyll and carotenoid contents with TiO2 application. We added:

‘Also, application of TiO2 acts as an anti-stress agent and induces an oxidation-reduction effect. In stressful conditions, application of TiO2 protects the chloroplast through activating antioxidant enzymes such as superoxide dismutase, peroxidase and catalase [17]. Chaudhary and Singh [15] noted that TiO2 nanoparticles could stabilize the integrality of chloroplast membrane and protect the chloroplasts against stressful conditions.’

‘Furthermore, TiO2 enhance nutrient accessibility by regulating enzymes activity involved in N metabolisms including nitrate reductase, glutamine synthase and etc. [31].’

  1. The paper contains a large number of grammar and syntax errors. Therefore, it has to be rewritten according with the scientific paper requirements.

Reply: We asked several colleagues who are skilled authors of English language papers to check the grammatically errors.

Reviewer 3 Report

Response to plants-1756636 Co-Application of TiO2 Nanoparticles and Arbuscular Mycorrhizal Fungi Improves Essential Oil Quantity and Quality of Sage (Salvia officinalis L.) in Drought Stress Conditions

Ostadi et al. investigated the influence of different irrigation regimes and fertilizer sources on the essential oil content and its quality of sage (Salvia officinalis L.). The irrigation treatments were 25, 50, and 75% maximum allowable depletion (MAD) percentage of the soil available water as non-stress (MAD25), moderate (MAD50) and severe (MAD75) water stress, respectively. Different subgroups including a no-fertilizer control, TiO2 nanoparticle (100 mg L-1), AMF inoculation, and co-addition of AMF and TiO2 were used. Moderate and severe drought stress decreased sage dry matter yield (DMY) by 30 and 65%, respectively. In contrast, application of TiO2 + AMF increased DMY and water use efficiency (WUE) by 35 and 35% compared to the no-fertilizer control. The highest EO content, yield and main EO constituent of sage (cis-thujone) was obtained in MAD50 fertilized with TiO2 + AMF. In addition, the net income index increased by 44, 47 and 76% with application of TiO2 nanoparticles, AMF and TiO2 nanoparticles + AMF, respectively.

The general idea of the manuscript is of potential interest and the study was well conducted but the authors should clarify some important issues, as follows:

1.      In the Introduction, please explain why the authors used TIO2 for minimizing the effects of different stress conditions in crop production and its quality. Please add information about the previous effect of TiO2 on essential oil content, nutrient concentration, chlorophyll and carotenoid content in other plants. What are the mechanisms involved? How do you explain these increases under the action of TiO2?

2.       Please add in Material and Methods information about the characterization of TiO2 nanoparticles used (size, zeta potential, method of preparation).

3.      Table 1, for example, does not show that there are significant differences between the effect of TiO2 and TiO2 + AFM compared to AFM. In my opinion, the best results are obtained with AFM or AFM + TiO2 but also thanks to AFM. Please explain why. Which are the mechanisms involved in TiO2 action on relative water content of sage, on essential oil content, water used efficiency, chlorophyll and carotenoid content?

4.      The paper contains a large number of grammar and syntax errors. Therefore, it has to be rewritten according with the scientific paper requirements.

Author Response

#Reviewer3

General comments

  1. The English grammar and style should be checked throughout the manuscript.

Reply: We asked several colleagues who are skilled authors of English language papers to check the grammatically errors.

  1. The authors should avoid using pronouns such as “we”, “our” and “us” in the text.

Reply: Done.

  1. Given that the study presents a long list of abbreviations, I suggest adding a “glossary” table at the end of the paper as it will aid the readers to learn about the concepts/terms that they are about to study.

Reply: The mentioned tables was added in the end of paper as Table 7.

Abstract

  1. The authors should add 1-2 sentences about the background of the study at the beginning of the Abstract section.

Reply: The background of study was added in first lines of abstract section: ‘Drought stress is known as a major yield-limiting factor in crop production that threatens food security worldwide. Arbuscular mycorrhizae fungi (AMF) and titanium dioxide (TiO2) have both shown to alleviate the effects of drought stress on plants but information about their co-addition to minimize drought stress is scant.’

  1. The authors should mention the main aim of the study before addressing the applied methods.

Reply: The main objective of study was added in this section: ‘Arbuscular mycorrhizae fungi (AMF) and titanium dioxide (TiO2) have both shown to alleviate the effects of drought stress on plants but information about their co-addition to minimize drought stress is scant.’

  1. I miss more emphasis on the main significance of this study in Abstract. I suggest highlighting the main significance of the study in 1-2 sentences.

Reply: The corrections were made.

  1. The authors should mention a few policy implications after the main recommendation based of results at the end of the Abstract in 1-2 sentences.

Reply: We added: Future policy discussions should focus on incentivizing growers to replace synthetic fertilizers with proven nano and biofertilizers to reduce environmental footprints and enhance the sustainability of sage production especially in drought conditions.

  1. The authors should avoid repeating keywords already exists in the title (e.g. drought stress). The authors should replace them with new relevant words in the text.

Reply: Done.

Introduction

  1. The Introduction section should be enriched by adding and citing several recent references (i.e. 2017- 2021). Also, the old references (1970-2005) should be replaced with the recent ones in the Introduction section as well as the whole manuscript.

Reply: The all references that used in introduction and discussion section was new and published between 2017-2022. However, based on the reviewer comments, the old references were deleted and new references and new findings of studies was added to text. The originality old references (1970-2005) that used in this study was related to measurement methods of AMF root colonization, nutrient’s concentration and also chlorophyll and carotenoid content. We added new references:

  1. Khater, R.M.R.; Sabry, R.M.; Pistelli, L.; Abd-Elgawad, A.M.; Soufan, W.; El-Gendy, A.N.G. Effect of compost and titanium dioxide application on the vegetative yield and essential oil composition of coriander. Sustain. 2022, 14, 1-11. https://doi.org/10.3390/su14010322.
  2. Shabbir, A.; Khan, M.M.A.; Ahmad, B.; Sadiq, Y.; Jaleel, H.; Uddin, M. Efficacy of TiO2 nanoparticles in enhancing the photosynthesis, essential oil and khusimol biosynthesis in vetiveria zizanioides Nash. Photosynthetica. 2019, 57, 599-606.https://doi.org/10.32615/ps.2019.071.
  3. El-Saadony, M.T.; ALmoshadak, A.S.; Shafi, M.E.; Albaqami, N.M.; Saad, A.M.; El-Tahan, A.M.; Desoky, E.S.M.; Elnahal, A.S.M.; Almakas, A.; Abd El-Mageed, T.A.; Taha, A.E.; Elrys, A.S.; Helmy, A.M. Vital roles of sustainable nano-fertilizers in improving plant quality and quantity-an updated review. Saudi J. Biol. Sci. 2021. 28, 7349–7359. https://doi.org/10.1016/j.sjbs.2021.08.032.
  4. Shalaby, T.A.; Bayoumi, Y.; Eid, Y.; Elbasiouny, H.; Elbehiry, F.; Prokisch, J.; El-Ramady, H.; Ling, W. Can nanofertilizers mitigate multiple environmental stresses for higher crop productivity? Sustain. 2022, 14, 1-22. https://doi.org/10.3390/su14063480.
  5. Shah, T.; Latif, S.; Saeed, F.; Ali, I.; Ullah, S.; Abdullah, A.A.; Jan, S.; Ahmad, P. Seed priming with titanium dioxide nanoparticles enhances seed vigor, leaf water status, and antioxidant enzyme activities in maize (Zea mays) under salinity stress. J. King Saud Univ. Sci. 2021, 33, 101207. https://doi.org/https://doi.org/10.1016/j.jksus.2020.10.004.
  6. Lines 76-78 in page 2: “Previous studies (e.g., ….) reported the positive effects of TiO2 nanoparticles on the improvement of plant performance, especially medicinal and aromatic plant, in stressful conditions.Ë® The authors should mention some of the previous studies in the underlined part of the sentence.

Reply: We added some previous study reports about application of TiO2 nanoparticles enhance essential oil quantity and its quality under drought stress conditions in introduction section. WE added:

‘Among different nanoparticles, TiO2 nanoparticles have positive impacts on the plant growth such as increasing nutrient uptakes, improving chlorophyll content and promote light capture in chlorophylls (a and b), regulation of important enzymes activities such as glutamine synthase, glutamate dehydrogenase and also nitrate reductase, chemical fixation of dinitrogen (N2) in the air, electron transfer activities, improving carbon dioxide (CO2) assimilation and photosynthetic activities [15, 16]. In addition, TiO2, as an anti-stress agent, decrease the negative effects of environmental stress by increasing the antioxidant enzymes activity [17].’

‘Previous studies reported the positive effects of TiO2 nanoparticles on the improvement of plant performance, especially medicinal and aromatic plant, in stressful conditions. Ah-mad et al. [18] noted that application of TiO2 nanoparticles increased the EO quantity and yield by 39 and 105% in peppermint seedlings. Gohari et al. [17] noted that application of TiO2 nanoparticles improve EO quantity and quality of Dracocephalum moldavica L. in stressful conditions.’

‘Also, Khater et al. [21] report ed that the foliar application of TiO2 nanoparticles im-proved EO quality of coriander through increasing the main EO constituents such as linalool.’

‘Shabbir et al. [22] reported that the application of TiO2 nanoparticles enhanced dry matter yield of vetiver (Vetiveria zizanioides L. Nash) as well as physiological characteristics such as total chlorophyll content, net photosynthetic rate, intercellular CO2 concentration, carbonic anhydrase and nitrate reductase activity. Also, the authors noted that the EO content and yield and also khusimol (main active constituent of EO) content increased by 23.6, 55.1 and 24.5% with application of TiO2 nanoparticles.’

  1. In the introduction section, the authors should add more explanations about the status and the use of sage in the world and in Iran.

Reply: more details of sage plants including global production per year in world and usage of this plant in different industries was added in text. However, there are no specific statistics about the cultivation area of this plant in Iran. We added:

‘The Global production of sage in world is estimated to 50-100 tons per year. Sage contains therapeutically effective compounds and has been used in the treatment of 60 diseases such as skin diseases, bronchitis, mouth and throat inflammations, digestive and circulation disturbances, cough, and other diseases. The sage essential oils (EO) could provide antioxidant, antimicrobial, antimutagenic, anti-cholinesterase, inflammatory, antibiotic benefits’

  1. In the Introduction section, there should be a paragraph discussing the global novelty of the study comparing with previous studies. This is very important to first identify the gap in the previous studies, and then highlight how the current study is going to fill it.

Reply: We revised the introduction section and add the novelty of this study. Also, we added the results of previous studies about the role of nanoparticles and biofertilizers in increasing plants productivity and its quality based on the previous comments. We added:

‘In addition, the lower nutrient use efficiency in drought stress conditions forces farmers to use more chemical inputs. A large portion of chemical fertilizers are lost and also become unavailable to plants in stressful conditions. For instance, 40-70% of N, 80-90% of P and 50-90% of K are lost and/or fixed in soils, resulting in economic losses [12, 13]. Two strategies to alleviate drought stress effects on plant production and quality are represented by harnessing advances in nanotechnology such as application of TiO2 and inoculating plants with biofertilizers such as arbuscular mycorrhizae Fungi (AMF).’

‘Integrating multiple strategies has proven to be more effective in minimizing the effects of different stress conditions in crop production and its quality [29]. Literature is scant on co-addition of TiO2 and AMF especially on sage. Therefore, the study was aimed to investigate the effectiveness of treatment with AMF and TiO2 nanoparticles on sage morphology, physiology, yield, and EO quality under drought conditions and investigate the economic incomes of each management practices to find the most environmentally friendly and economically viable options for growers.’

Methodology

  1. Why the “4. Materials and MethodsË® is placed after Discussion section? The Methodoly should immediately appear after Introduction section. Therefore, the authors should move this section after Introduction section as “2. Materials and MethodsË®

Reply: In the template of Plants journal the ‘materials and methods section’ added in the fourth part and after the discussion.

  1. The first sub-section of the Methodology section should be 3.1. Study area.

Reply: Done.

  1. The location of the study area is not specified exactly. In the “3.1. Study area” subsection, the authors should discuss the descriptions of the study area exactly.

Reply: Geographical details of experimental site were added to the text: ‘The present study was conducted during two growing years (2019 and 2020) at the re-search farm of Maragheh University, Iran (longitude 46°16' E, latitude 37°23' N, altitude 1532 m).’

  1. Also, the authors should add a credible map of the study domain.

Reply: The aerial photo of experimental area was added in text as Figure 7.

  1. All equations should have number and cite in the main text.

Reply: Done.

  1. Equation 52 seems copy/paste and the authors should write it.

Reply: The corrections were made.

Results

  1. Lines 139 in page 4: The caption of the Table 1 seems have different format.

Reply: Checked.

  1. In the Results section, the resolution of the Figure 6 should be enhanced.

Reply: Done.

  1. Overall, the Results section is well-developed and explanations of the findings are satisfying.

Reply: Thanks.

Discussion

  1. In this section, inputs need to be supported with sufficient and relevant references. The findings and their implications should be discussed in the broadest context possible and limitations of the work. Moreover, the authors should compare/contrast their findings with similar studies (2017-2022).

Reply: This section was revised based on the reviewer comments and use the new findings of studies for comparing of obtained results to published results. For example:

‘Alike, Amani Machiani et al. [26] reported that the thyme (Thymus vulgaris L.) DMY reduced in moderate and sever water scarcity due to the decreasing of macro- and micro-nutrients uptake in drought conditions.’

‘Similarly, Govahi et al. [38] noted that the maximum EO content of sage enhanced by 109 and 84% in moderate and severe drought stress conditions, respectively.’

‘Hazzoumi et al. [47] reported that the application of AMF in water stress conditions enhances the RWC content of basil (Ocimum gratissimum L) leaves.’

Conclusion

  1. In the Conclusion section, the authors should discuss the main implication of the findings.

Reply: The main obtained results was added in this section.

  1. The future research directions should be discussed at the end of this section as well.

Reply: We added: The future researches should be focused on the effects of different nanoparticles as well as different AMF species for alleviating drought stress impacts and improving EO quantity and quality of medicinal and aromatic plants.

Round 2

Reviewer 3 Report

The authors have satisfactory answered to all questions mentioned.